# The Arsenal of *Leptospira* Species against Oxidants

**DOI:** 10.3390/antiox12061273

**Published:** 2023-06-14

**Authors:** Samuel G. Huete, Nadia Benaroudj

**Affiliations:** Institut Pasteur, Université Paris Cité, Biologie des Spirochètes, CNRS UMR 6047, F-75015 Paris, France

**Keywords:** spirochetes, *Leptospira*, ROS, catalase, peroxidase, superoxide dismutase, PerR, OxyR, SoxRS, OhrR

## Abstract

Reactive oxygen species (ROS) are byproducts of oxygen metabolism produced by virtually all organisms living in an oxic environment. ROS are also produced by phagocytic cells in response to microorganism invasion. These highly reactive molecules can damage cellular constituents (proteins, DNA, and lipids) and exhibit antimicrobial activities when present in sufficient amount. Consequently, microorganisms have evolved defense mechanisms to counteract ROS-induced oxidative damage. *Leptospira* are diderm bacteria form the *Spirochaetes* phylum. This genus is diverse, encompassing both free-living non-pathogenic bacteria as well as pathogenic species responsible for leptospirosis, a widespread zoonotic disease. All leptospires are exposed to ROS in the environment, but only pathogenic species are well-equipped to sustain the oxidative stress encountered inside their hosts during infection. Importantly, this ability plays a pivotal role in *Leptospira* virulence. In this review, we describe the ROS encountered by *Leptospira* in their different ecological niches and outline the repertoire of defense mechanisms identified so far in these bacteria to scavenge deadly ROS. We also review the mechanisms controlling the expression of these antioxidants systems and recent advances in understanding the contribution of Peroxide Stress Regulators in *Leptospira* adaptation to oxidative stress.

## 1. Introduction

*Leptospira* are aerobic diderm bacteria of the *Spirochaetes* phylum. They are thin, helical-shaped bacteria with a periplasmic endoflagellum, and highly motile organisms. *Leptospira* genus includes free-living bacteria that can be found in aqueous environments, as well as pathogenic species that can infect and colonize mammalian hosts. They are the causative agent of leptospirosis, a widespread zoonosis [1]. Dissemination of pathogenic *Leptospira* in the environment depends on reservoir hosts, mainly rodents, which are asymptomatic carriers. Leptospires chronically colonize the proximal renal tubules of these mammals and are excreted in the environment by their urine. Leptospires are transmitted to other animals and humans mostly by exposure to soils and water contaminated by the bacteria. *Leptospira* penetrate host organisms by abraded skins and mucosa, and they rapidly disseminate via the bloodstream to numerous tissues and organs including kidneys, liver, and lungs. Their remarkable motility, their ability to resist complement killing and avoid recognition by phagocytic cells, as well as a great adaptation to the host environment contribute to a rapid host colonization [1]. The zoonotic transmission also implies that pathogenic *Leptospira* are able to survive, presumably for a long time, in soils and water.

Leptospirosis symptoms range from a mild flu-like febrile state to more severe and fatal cases leading to hemorrhages and multiple organ failure. Patients with acute leptospirosis experience jaundice and renal failure (Weil’s disease), severe pulmonary hemorrhages syndrome (SPHS), and meningitis [2]. The lack of specificity of the clinical manifestations associated with leptospirosis complicates the diagnosis of this disease which is often mistaken with other illnesses and consequently underdiagnosed. One million cases of severe leptospirosis are estimated annually worldwide with about 60,000 deaths [3]. However, fatality among patients experiencing SPHS can reach up to 70% [4]. The highest leptospirosis incidence is generally observed among an impoverished population of tropical regions, classifying this disease as neglected zoonosis [5]. Global climate change, possible increase in extreme weather events (including flooding), and the multiplication of slum habitations with poor sanitization are predicted to augment leptospirosis incidence [6]. Interestingly, reported cases of leptospirosis are increasing also in developed countries under temperate climates [7]. Such changes in leptospirosis epidemiology should improve the attention this disease merits, which can be considered as an emerging zoonosis.

*Leptospira* are understudied bacteria as their investigation is impaired by several technical obstacles. First, *Leptospira* are fastidious and relatively slow-growing organisms in laboratory conditions. The doubling time of pathogenic species in liquid medium is about 20–24 h and colonies appear on agar plate after one month of incubation. In addition, gene inactivation by allelic exchange is feasible but very ineffective in pathogenic species. Recent development of technologies for gene silencing [8,9,10] should facilitate identification of virulence factors but our understanding of *Leptospira* pathogenesis mechanisms is far from being completely understood.

Both saprophytes and pathogenic *Leptospira* species, as any aerobic bacteria, are exposed to ROS produced endogenously through their own metabolism or generated in the outside environment. Pathogenic species are further confronted by deadly ROS produced by the host as part of the innate immune defenses. These species are thus well-equipped to withstand the oxidative stress encountered when infecting their hosts and their ability to tolerate oxidative stress plays a pivotal role in their virulence.

Microbial adaptation to oxidative stress has been studied for several decades and is well-documented in model bacteria such as *Escherichia coli*. Numerous excellent reviews have been written on how these model bacteria tolerate oxidative stress [11,12,13], but this subject is rarely addressed in atypical and poorly characterized bacteria such as *Leptospira*. Here, we review our current knowledge on the mechanisms used by pathogenic *Leptospira* to withstand different ROS and how these pathways are regulated.

## 2. Reactive Oxygen Species: What, When and How?

The great oxidation event (GOE), that occurred between 2.5 and 2 billion years ago, marked the transition from an anoxic to an oxygen-rich environment [14]. As the atmosphere became oxidizing, living organisms evolved to use oxygen and to cope with its metabolic toxic byproducts.

ROS, including superoxide (O_2_^•−^), hydrogen peroxide (H_2_O_2_), hypochlorous acid (HOCl), organic hydroperoxides (ROOH), and hydroxyl radical (^•^OH), are molecules produced through electron transfer to oxygen and are harmful when present in excess. Oxygen possesses two unpaired electrons in separate orbitals and is susceptible to reduction. ROS are generated through several consecutive monovalent electron transfers (Figure 1). Superoxide is produced after the transfer of a single electron to molecular oxygen. A second electron transfer to superoxide gives rise to H_2_O_2_ which can be subsequently reduced into highly reactive hydroxyl radical molecules.

ROS have different reactivities. Hydroxyl radicals are by far the most reactive ROS. They can interact with almost all biomolecules, including proteins, DNAs and lipids, and are believed to have a large spectrum of targets. On the contrary, O_2_^•−^ and H_2_O_2_ are less reactive and perform more specific oxidation. It should be noted though that as O_2_^•−^ and H_2_O_2_ are precursors of ^•^OH, it is difficult to distinguish oxidative damage due to O_2_^•−^ and H_2_O_2_ from those occurring consequently to the accumulation of ^•^OH. Using bacterial mutants that accumulate O_2_^•−^ and H_2_O_2_, it was shown that ROS induce DNA mutations [15,16]. The deoxyribose and base moieties can be oxidized and 8-oxo 7,8-dihydroguanosine is the major nucleotide modification observed in DNA in the presence of oxidants [17]. All RNA molecules (rRNA, tRNA, and mRNA) are also vulnerable to oxidation by ROS and their oxidation reduces translation rate [18,19]. Strand breaks are also observed when DNA and RNA are oxidized.

Protein backbones and lateral chains can be oxidized by ROS, which results in increased carbonyl content in proteins, protein–protein crosslinking and protein fragmentation [20,21]. Sulfur-containing amino acids are particularly susceptible to oxidative damage. Disulfide bridges and cysteic acids are formed when cysteines are oxidized; methionine sulfoxide and sulfone are products of methionine oxidation [22].

H_2_O_2_ interacts with iron located in iron-containing proteins leading to their inactivation and fueling the Fenton reaction, the second step of the Haber–Weiss reaction (Figure 1, reaction 2) [23]. In addition, superoxide can extract iron from iron–sulfur clusters contained in bacterial enzymes, leading to enzyme inactivation [24,25].

### 2.1. Endogenous Sources of ROS in Bacteria

In bacteria, several flavin- or quinones-containing enzymes from the respiratory chain are potent electron donors that enhance O_2_^•−^ production from electron leaking [26,27]. In addition, superoxide dismutase and reductase enzymes catalyze the dismutation and reduction in superoxide into H_2_O_2_ (Figure 1). Iron released from haem group, iron–sulfur clusters, or mononuclear iron enzymes will be engaged in the Fenton reaction that generates hydroxyl radicals from H_2_O_2_ (Figure 1).

The intracellular concentrations of H_2_O_2_ and O_2_^•−^ in *E. coli* have been estimated to 20–50 nM and 200 nM, respectively [11,28]. Inactivation of several enzymes detoxifying H_2_O_2_ can increase the amount of the steady state level of this ROS to 2 μM, resulting in growth defect [28]. One important question is to determine to what extent the level of ROS produced physiologically is harmful for the bacteria or whether ROS might rather act as signaling molecules when present at a tolerable amount as observed in eukaryotes.

### 2.2. Exogenous Sources of ROS Encountered by Bacteria

In addition to ROS produced endogenously, living organisms are exposed to ROS produced in the outside environment by ionizing radiation, UV light, and a large variety of chemicals (e.g., paraquat/methyl viologen) [29,30] (Figure 2).

Commensal, symbiotic, and pathogen bacteria are exposed to ROS produced inside their eukaryotic hosts. Macrophages and neutrophiles contain an inducible NADPH-dependent oxidase (Phox) that produces superoxide in response to the detection of invading pathogens [31,32]. As described above, superoxide can be dismutated to H_2_O_2_ through a SOD-catalyzed reaction and H_2_O_2_ is further processed in hydroxyl radicals. In neutrophiles, H_2_O_2_ reacts with chloride ions to generate hypochlorous acid through a reaction catalyzed by myeloperoxidase (MPO) (Figure 1) [33]. In addition, superoxide can interact with nitric oxide (NO^•^) produced in phagocytes by an inducible nitric oxide oxidase to form peroxynitrite (ONO_2_^−^). Of note, NO^•^ and ONO_2_^−^ are reactive nitrogen species (RNS) that are as oxidative as ROS [34].

Phagocytes are not the only eukaryotic cells that produce oxidants. Indeed, NADPH oxidases (Nox and Duox) are also expressed in a variety of other cell types including epithelium and endothelium. As a matter of fact, H_2_O_2_ is detected in lungs [35], kidneys [36], ocular tissues [37], as well as in fluids such as urine [38,39] and blood [37].

Oxidative respiration in mitochondria, present in phagocytic cells or in tissues, is another source of ROS in eukaryotes [40]. Electrons are shuttled through the electron transport chain complexes I–IV located in the mitochondrial inner membrane. Electron leaking from the electron transport chain and their improper transfer to oxygen results in superoxide production. Production of ROS in mitochondrial matrix is stimulated by pathogen-associated molecular patterns (PAMPs) [41].

Attempts to estimate the concentration of oxidants inside a phagosome suggested that 2–50 μM O_2_^•−^ and 1–4 μM H_2_O_2_ are produced by macrophages [42] and neutrophils would contain 25 μM O_2_^•−^ [43]. An estimated concentration of 10–200 pM O_2_^•−^ is present in the mitochondrial matrix [40]. ROS produced by the host have bactericidal activities, but they also participate in the activation of other antibacterial pathways of the innate immunity including generation of extracellular traps by the neutrophiles (NETosis) and production of pro-inflammatory molecules.

## 3. Are *Leptospira* Exposed to ROS in Their Different Ecological Niches?

*Leptospira* species are aerobic bacteria that possess a pathway for oxidative phosphorylation (Figure 2). Genes encoding putative NADH dehydrogenase (LIMLP_00745-00750, LIMLP_03705-03760), succinate dehydrogenase (LIMLP_09980-09985), cytochrome C oxidases (LIMLP_01080-01085, LIMLP_01375, LIMLP_01100-01110), and a F1 ATP-synthase complex (LIMLP_06045-06080) are annotated in the pathogen *L. interrogans* genomes (serovar Manilae strain UP-MMC-NIID LP) [44]. The endogenous production of ROS has never been measured in *Leptospira*, but it is very likely that these bacteria also produce oxidants as a result of electron leaking from the respiratory chain.

Most *Leptospira* species are free-living organisms that can be isolated from soils, freshwaters, stagnant waters, sewage, and even drinking water [45,46,47,48]. Surface soils and waters are well-oxygenated and exposed to sunlight energy. All these aquatic and soil microcosms contain ROS arising from abiotic reactions and from soil microbiome and plants [49,50]. Studies establishing a correlation between *Leptospira* survival and soil composition are scarce. Lall et al. [51] have shown that the presence of *Leptospira* correlates with the presence of iron, copper, and manganese, metals that exacerbate ROS production. Even though there are no studies that have measured the level of ROS in microcosmos-containing *Leptospira*, it is reasonable to conjecture that leptospires are exposed to oxygen byproducts in their free-living lifestyle (Figure 2). Furthermore, *Leptospira* have been shown to form aggregates [52] and biofilms [45,53] in aquatic environments and to what extent these lifestyles protect them from oxidative stress, remains to be evaluated.

Pathogenic *Leptospira* species are mainly extracellular bacteria that escape phagocytosis by host immune cells. They colonize different tissues including lung, kidney, and liver, that are highly oxygenated and therefore ROS producers [54] (Figure 2). Several studies have evaluated whether leptospirosis correlated with an increased oxidative stress in the host. Blood of leptospirosis patients contains a higher amount of ROS and NO^•^, and a lower amount of gluthathione (GSH), an anti-oxidant, than healthy individuals [55,56]. A correlation between leptospirosis and oxidative stress has also been observed in infected animals [57,58].

A higher number of circulating neutrophiles has been reported during human leptospirosis [59] and NETosis has been implicated in leptospires killing [60,61], but pathogenic *Leptospira* remain at the surface of neutrophiles without triggering ROS production by MPO [62]. Interestingly, the non-pathogenic strain, *L. biflexa*, exhibit a different behavior in the same condition as they were found inside phagocytic vacuoles of neutrophiles and they triggered ROS production [62]. This might indicate that pathogenic *Leptospira* species are able to efficiently detoxify extracellular ROS whereas saprophytes do not.

Even though pathogenic leptospires escape phagocytosis by host immune cells, their presence inside macrophages have been demonstrated [63,64,65,66,67]. However, only few studies have shown that macrophages infected with leptospires produced a higher amount of ROS [68,69].

Establishing a correlation between ROS production and infection by leptospires does not necessarily prove that the level of ROS produced by the host threatens *Leptospira* survival. Instead, the demonstration that these pathogens are confronted to deadly ROS during infection is founded on a study by Eshghi et al. [70], showing that catalase mutants of *L. interrogans* are avirulent. There is, therefore, compelling evidence that pathogenic *Leptospira* are exposed to a host-triggered oxidative stress during infection and that their ability to withstand this stress is important for their virulence.

## 4. Defenses against ROS

Defenses against ROS include non-enzymatic low molecular weight molecules which act as thiol-redox buffers, detoxification enzymes that breakdown ROS and repair machineries that reverse oxidative damage to molecules.

### 4.1. Catalase and Peroxidases

Catalases and peroxidases are oxidoreductases that catalyze H_2_O_2_ breakdown. Catalase degrades two molecules of H_2_O_2_ into H_2_O and O_2_ through a dismutation reaction (Figure 3). They are classified in three main groups: monofunctional catalases, bifunctional catalase-peroxidases, and manganese-containing catalases (reviewed in [71]). Mono- and bifunctional catalases use heme prosthetic group as a cofactor for oxidoreduction whereas manganese-containing catalases are non-heme enzymes that use manganese ions for oxidoreduction. In addition to classical catalase activity, bifunctional catalases also bear a peroxidase activity.

Pathogenic *Leptospira* species express a monofunctional heme-containing catalase (encoded by LIMLP_10145, *katE*, in *L. interrogans*) (Figure 4).

Far ahead the availability of *Leptospira* genomes, two studies demonstrated that a catalase activity can be detected in several strains of pathogenic *Leptospira* [78,79]. This catalase is localized in the periplasm of *Leptospira* and its inactivation showed that not only this catalase is the major H_2_O_2_ detoxification enzyme but its activity is required for virulence in *L. interrogans* [70]. *KatE* is the second gene of an operon with LIMLP_10150 which encodes an ankyrin repeat-containing protein [80]. This operon is up-regulated in the presence of H_2_O_2_ [80], at 37 °C [81,82], and when *Leptospira* are cultivated within a dialysis membrane chamber implanted in rat peritoneal cavities [83,84]. The function of the LIMLP_10150-encoding protein is unknown but similar ankyrin repeat-containing proteins have been shown to interact with and promote the catalase activity in other bacteria [85,86,87]. It has been proposed that the ankyrin domain-containing protein stabilizes the catalase in a proper conformation or orientation to promote H_2_O_2_ entry in the active site [85] or heme binding [86]. Corin et al. [88] showed that saprophytes such as *L. biflexa* did not exhibit a detectable catalase activity even though genome sequencing showed the presence of a bifunctional catalase-peroxidase encoded by *katG* (LEPBIa2495) in saprophytes.

Peroxidases reduce H_2_O_2_ into H_2_O using an electron donor (thioredoxin, NADH, and NADPH) (Figure 3) (reviewed in [89]). They have a larger spectrum of substrates than catalases and they also reduce organic hydroperoxides (RO_2_H) into their corresponding alcohols (ROH). They are classified into two families, i.e., thiol peroxidases and cytochrome C peroxidases. Thiol peroxidases reduce H_2_O_2_ and organic hydroperoxides by a mechanism involving the oxidation of catalytic cysteine residues (formation of a cysteine sulfenic acid and disulfide bond). They are further subclassified in different families according to the disulfide reductase system that regenerates the oxidized peroxidase into an active reduced peroxidase. For instance, the bacterial alkylhydroperoxidase, AhpC, most often uses AhpF, a NADH:disulfide oxidoreductase flavoprotein, as reductase. However, in some bacteria (*Mycobacterium tuberculosis* and *Helicobacter pylori*), AhpF is absent and AhpC is reduced by a thioredoxin (Trx) and a thioredoxin reductase (TrxR) (Figure 3). Both AhpF and the TrxR are finally reduced by NADPH.

An AhpC-encoding ORF has been identified in *L. interrogans* (LIMLP_05955, AhpC1) but this species belongs to the category of bacteria that do not possess AhpF whereas *L. biflexa* genome encodes two AhpC paralogs (AhpC1, LEPBIa1358 and AhpC2, and LEPBIa3009) and one AhpF (LEPBIa3008) (Figure 4). A role of *L. interrogans* AhpC1 in degrading peroxides was first shown in vitro using a recombinant protein and by overexpressing leptospiral *ahpC1* in *E. coli* [90]. The Trx/TrxR system that reduces AhpC1 has not been identified in *L. interrogans*; however, the system encoded by *trxA* (thioredoxin LIMLP_09870) and *trxRB* (thioredoxin reductase LIMLP_07165) was shown to have a reducing activity in vitro in the presence of NADPH, suggesting that its function can be to reduce AhpC1 in vivo. Inactivation of *ahpC1* in *L. interrogans* impaired their ability to grow in the presence of paraquat, a superoxide-generating compound, but it did not in the presence of H_2_O_2_ [80]. This was surprising as *ahpC1*, as well as *trxA* and *trxRB*, are up-regulated in the presence of H_2_O_2_ [80]. This can be explained by the fact that H_2_O_2_ produced exogenously would be mainly reduced by the periplasmic catalase present in pathogenic *Leptospira* before it can reach the cytoplasm. In this scenario, AhpC1 would reduce H_2_O_2_ present in *Leptospira* cytoplasm, as arising for instance by superoxide reduction.

The bacterioferritin comigratory protein (BCP), thiol peroxidase (Tpx), and Glutathione peroxidase (Gpx) have peroxidase activities demonstrated in vitro against H_2_O_2_ and organic peroxide, even though the physiological reducing systems devoted to their regeneration is not always known [91,92,93,94]. These enzymes contribute to some extent to peroxide breakdown in vivo in several bacteria [91,93,95,96,97,98,99] but in which exact particular condition they fulfill their function is a matter of discussion [100]. *Leptospira* genomes contain ORFs annotated as BCP (LIMLP_18310), Tpx (LIMLP_03630), and Gpx (LIMLP_13255, LIMLP_17550). Neither the enzymatic activities nor the functions of the proteins encoded by these ORFs have been characterized. However, the up-regulation of BCP and Gpx-encoding ORFs in the presence of H_2_O_2_ in *L. interrogans* suggests a role in defense against ROS [80].

The second main family of peroxidases are cytochrome C peroxidases (CCP) that catalyze the reduction in H_2_O_2_ using electrons from cytochrome C. In bacteria, CCPs are dimeric proteins where each monomer contains two, or more rarely three, heme binding domains (reviewed in [101]). They are located in the periplasm and can be soluble or anchored to inner or outer membranes. Electrons are first transferred from cytochrome C to the high potential (hp) heme-binding domain, at the C-terminus of the protein. Then, the electrons are transferred from the reduced hp heme-binding domain to the low potential (lp) heme-binding domain located at the N-terminus of the protein. The lp heme-binding domain contains the binding site for H_2_O_2_. After two consecutive electron transfers from the hp heme-binding domain, H_2_O_2_ is reduced into two molecules of water (Figure 3d). One role of CCPs is to detoxify exogenous H_2_O_2_ that have accumulated in the periplasm. Recently, Khademian and Imlay [102] proposed that CCP can also be a reductase allowing H_2_O_2_ to act as final electron acceptor under anaerobic condition.

*L. interrogans* and *biflexa* each encode 4 ORFs annotated as CCP (LIMLP_02795, LIMLP_04655, LIMLP_05260 and LIMLP_14625; LEPBIa1208, LEPBIa2430, LEPBIa2855, and LEPBIa5260). However, two of them (LIMLP_05260 and LIMLP_14325; LEPBIa3120 and LEPBIa1208) are probably MauG-like proteins involved in the methylamine metabolism pathway, as suggested by the presence of a specific Tyr residue (Y338 in LIMLP_05260, Y291 in LIMLP_14325, Y351 in LEPBIa3120, Y339 in LEPBIa1208). The LIMLP_02795 ORF is dramatically up-regulated in the presence of H_2_O_2_ [80], which suggests a role of the CCP encoded by this ORF either in detoxifying H_2_O_2_ or using H_2_O_2_ as electron acceptor. Yet, such a role remains to be experimentally demonstrated either by a biochemical characterization of the protein or phenotypic studies of a mutant inactivated in LIMLP_02795.

### 4.2. Low Molecular Weight (LMW) Thiol Redox Buffers

The best characterized LMW thiol molecule is glutathione (GSH), a tripeptide composed of glutamate, cysteine, and glycine. The glutamate and cysteine are linked by a peptide bound between the gamma-carboxyl of glutamate and the amino group of cysteine. The synthesis of GSH is catalyzed by gamma-glutamylcysteine ligase (GCL) and glutathione synthetase (GS) in *E. coli*, and GSH can reach a concentration estimated in the millimolar range [103]. This tripeptide bears a reactive sulfhydryl group that can maintain the redox of cells [104]. GSH can be oxidized into a disulfide-bonded form (GSSG). GSH and GSSG form a redox pair acting as an electron donor and acceptor in redox reactions. One role of GSH is to regenerate the reduced state of thiol enzymes such as glutathione peroxidase, peroxiredoxin, and glutaredoxin. GSH is also a cofactor of protein disulfide isomerases (PDIs) which catalyze the isomerization and reduction in disulfides. During these reactions, GSH is oxidized into GSSG (Figure 3c). To recover the pool of GSH and restore the redox buffer of bacteria, GSSG is reduced into GSH by the glutathione reductase.

Cysteine residues are particularly prone to oxidation giving rise to disulfide bonds and sulfenic acid (Cys-SOH). These oxidations are reversible but overoxidation into sulfunic and sulfonic acids (Cys-SO_2_H and Cys-SO_3_H) irreversibly modifies and damages proteins. To avoid this, GSH can form transient disulfide bridges with cysteines in proteins exposed to ROS. This modification is known as S-glutathionylation and is catalyzed by the glutathione S transferase (GST). Glutaredoxins catalyze the deglutathionylation to restore reduced cysteines.

Many enzymes of the GSH synthesis and metabolism can be identified in *Leptospira* genomes. Indeed, LIMLP_08995 and LIMLP_ 08990 encode putative GCL and GS, respectively, in *L. interrogans*. GST-encoding ORFs are also annotated in *Leptospira* genomes (LIMLP_02530, LIMLP_06655, and LIMLP_13670), so are glutaredoxins (LIMLP_08980, LIMLP_08985), GSH hydrolase (LIMLP_09000) that catalyzes GSH breakdown into cysteinylglycine and glutamate, and glutathione peroxidases (LIMLP_13255, LIMLP_17550). Importantly, GCL, GS, and glutaredoxin activities can be measured in *L. interrogans* lysates, demonstrating the existence of a GSH synthesis and metabolism in these bacteria [105]. Despite the absence of an ORF encoding a GR, such activity can be also detected in *Leptospira* lysates, but the redox system responsible for reducing GSSG is unknown. LIMLP_08950 is the phylogenetically closest ORF that can fulfill a GR function.

In Sasoni et al. [105], a much lower GSH content was detected in *L. interrogans* than in *E. coli* (60 and 1300 fmol for 10^7^ *L. interrogans* and *E. coli*, respectively). The exact contribution of GSH in maintaining *Leptospira* redox is thus unclear. It is worth mentioning that the two glutaredoxins and one GST (LIMLP_13670) are up-regulated in *Leptospira* upon oxidative stress [80]. Another evidence of a participation of glutathionylation and deglutathionylation in the oxidative stress response of *Leptospira* is the finding that the two leptospiral glutaredoxins (encoded by LIC11809 and LIC11810, the homologs of LIMLP_08980 and LIMLP_08985 in *L. interrogans* serovar Copenhageni strain Fiocruz L1-130, respectively) can complement the growth defect of *Saccharomyces cerevisiae* mutants lacking the endogenous glutaredoxin in the presence of H_2_O_2_ [106].

### 4.3. Superoxide Dismutase and Reductase

Superoxide is the precursor of many ROSs (including H_2_O_2_ and ^•^OH, see Figure 1) and defenses against superoxide are vital for many bacteria to survive. It is generally believed that all living organisms own systems for superoxide detoxification [107]. The most common enzymes in charge of superoxide removal are superoxide dismutase (SOD) and superoxide reductase (SOR). SOD are metalloenzymes that use a metal (Fe, Mn, Cu, Zn, or Ni) as electron acceptor and donor to catalyze the dismutation reaction of superoxide into H_2_O_2_ and O_2_ (Figure 5).

In the superoxide dismutation reaction, one molecule of superoxide is oxidized into O_2_ by transferring one electron to the SOD-bound metal. A second molecule of superoxide is then reduced into H_2_O_2_ by gaining electron from the reduced SOD, restoring the initial state of the enzyme with no external reducer [108]. SODs are divided into three families according to the metals used for catalysis: Fe/MnSODs (SodB/SodA), Cu/ZnSODs (SodC), and NiSODs [107]. Fe- and MnSODs are closely related. Some members of this family strictly depend on either Fe^2+^ or Mn^2+^, whereas others (the cambialistic SODs) can coordinate and be active with both metals. SODs are located in the cytosol (Fe/MnSOD, NiSOD) or in the periplasm (Cu/ZnSOD).

As opposed to the double reduction-oxidation mechanism catalyzed by SODs, iron-binding SOR enzymes only catalyze the reduction in the O_2_^•−^ radical into H_2_O_2_ (Figure 5). This is a fundamental difference with SODs since oxidized SORs need to be reduced back to their initial state by a reducer. Rubredoxins have been identified as electron donors for SORs in the anaerobes *Archaeoglobus fulgidus* [109] and *Desulfovibrio vulgaris* [110].

Most bacteria possess one or several isoforms of SODs and/or SORs. Anaerobes would preferentially contain SORs since these enzymes allow removal of superoxide without producing oxygen. Saprophytic *L. biflexa* encode a Fe/MnSOD (LEPBIa0027) that shares 71% identity with the *E. coli* SodB (Figure 4) and exhibits the amino acids required for Fe^2+^/Mn^2+^ coordination (H27, H74, D157, and H161). Biochemical characterization is needed to determine whether *L. biflexa* SodB uses preferentially Fe^2+^ or Mn^2+^ for catalyzing superoxide removal or whether the two metals are interchangeable.

Surprisingly, neither SOD nor SOR orthologs have been identified in pathogenic *Leptospira* spp. [111]. These observations are also supported by a study showing absence of detectable SOD activity in *L. interrogans* cultures as opposed to *L. biflexa* [112]. It is interesting to note that SOR-like enzymes have been described in other pathogenic spirochetes such as *Treponema* spp. (TP0823) [113] and a functional MnSOD has been described in *Borrelia* spp. (BB0153) whose role is essential for virulence [114,115]. Therefore, pathogenic *Leptospira* spp. remain an enigma concerning their mechanism of superoxide tolerance, if any. It should be noted that only very few bacterial species are known to lack any enzymatic superoxide removal mechanism, one of them being *Lactobacillus plantarum*, which uses manganese as a superoxide scavenger [116,117]. SOD/SOR-independent mechanism of superoxide scavenging relying on manganese has also been observed in *Neisseria gonorrhoeae* [118], but this has never been explored in pathogenic *Leptospira*.

## 5. Regulation of the Oxidative Stress Response

Defenses against accumulation of deadly ROS in bacteria are tightly regulated transcriptionally in order to maintain intracellular ROS homeostasis adapted to the environment the bacteria are facing. There are several transcriptional regulators involved in the regulation of the adaptive response to oxidative stress, including OxyR, PerR, OhrR, and SoxRS. All these regulators have in common an ability to sense the presence of ROS through amino-acid oxidation and to trigger the appropriate transcriptional response, i. e., “oxidative stress regulon”, that confers a better ability to survive under oxidative stress. Thus, ROS can also function in bacteria as signaling molecules to activate or derepress expression of target genes.

OxyR, PerR, OhrR, and SoxRS are not present in all bacteria but several of them can co-exist within the same bacterial species and their respective regulon may overlap to some extent.

### 5.1. OxyR

OxyR is a 34 kDa protein of the LysR transcriptional regulator family that self-associates into a tetramer (Figure 6). It is activated when an intramolecular disulfide bond is formed between two cysteine residues (C199 and C208, according to *E. coli* OxyR sequence) in the presence of H_2_O_2_ or upon modification of the redox status of bacteria [119,120]. This leads to a conformational switch resulting into a higher affinity for DNA and favoring thereby the interaction of the RNA polymerase with DNA [121]. Oxidation of OxyR is reversible and the disulfide bridge is reduced by the glutaredoxin and thioredoxin systems [120]. When bound to DNA, oxidized OxyR mostly activates the expression of genes encoding proteins involved in H_2_O_2_ removal (catalase, and alkyl hydroxyperoxidase), and in maintaining the thiol redox (glutathione oxidoreductase, glutaredoxin, thioredoxin, and thioredoxin reductase) [122,123]. OxyR regulon also encompasses genes involved in lowering the content of free iron (*fur*, *dps*, *yaaA*) in heme biosynthesis (*hemH*), and in iron–sulfur cluster assembly (*sufABCDE*, *sufS*) [124]. Similar to any canonical LysR transcriptional regulator, OxyR represses its own expression. Mutants inactivated in *oxyR* are generally more sensitive to H_2_O_2_ than the WT strain [125,126].

### 5.2. PerR

Peroxide stress regulator (PerR) belongs to the Fur transcriptional repressor family. PerR was first identified and described in *Bacillus subtilis* [127,128]. It is a dimer of two 17 kDa protomers with each protomer having an amino-terminal DNA binding domain and a carboxy-terminal dimerization domain. PerR has a regulatory metal binding site composed of three histidines and two aspartates (H37, H91, H93, D104, and D85 according to the PerR sequence in *B. subtilis*) located at the hinge between the two domains and that controls DNA binding. When the regulatory metal (Fe^2+^ or Mn^2+^) occupied the metal binding site, PerR adopts a conformation with a high affinity to DNA, leading to repression of PerR-controlled genes [129,130,131]. In the presence of H_2_O_2_, H37 and H91 in iron-bound PerR are oxidized into 2-oxohistidines [132]. This oxidation is mediated by hydroxyl radicals produced by H_2_O_2_ and the Fe^2+^ coordinated in the regulatory metal-binding through a Fenton reaction [132]. Therefore, Mn^2+^ can function as a surrogate regulatory metal for DNA binding, but not as a H_2_O_2_ sensing metal. PerR oxidation induces a conformational switch leading to PerR dissociation from DNA and derepression of the PerR regulon [132,133] (Figure 6).

PerR oxidation is irreversible and, in *B. subtilis*, oxidized PerR is degraded by Lon protease [134]. PerR generally regulates its own expression and the expression of oxidative stress-related genes. In *B. subtilis*, genes encoding catalase, AhpCF, and CCP are repressed by PerR as well as genes coding for MrgA, a Dps analog, Fur, and the heme biosynthesis machinery (*hemAXCDBL*) [128,135,136]. Inactivation of *perR* leads to a greater tolerance to H_2_O_2_ in many bacteria [128,137,138,139].

OxyR and PerR are evolutionarily distinct, but they are functional homologs as they both control genes involved in defenses against peroxide stress. OxyR mostly exists in Gram-negative bacteria whereas PerR is generally found in Gram-positive bacteria. There are exceptions as OxyR homologs can be found in some actinobacteria (*Mycobacterium leprae*, *Corynebacterium glutamicum*, *Streptomyces coelicolor*) [140] and PerR homologs exist in some proteobacteria [137]. OxyR and PerR rarely coexist within the same bacterial species; however, they can be both found in *Deinococcus radiodurans* and *Neisseria gonorrhoeae* [141].

### 5.3. OhrR

Organic hydroperoxide resistance regulator) (Ohrh) is a transcriptional repressor of the MarR family well-distributed in bacteria. Similar to PerR, OhrR binds DNA when in its reduced form, resulting in gene repression (Figure 6). It senses organic hydroperoxides (RO_2_H) through a single cysteine residue (C15 in *B. subtilis* OhrR) which is oxidized into cysteine sulfenic acid (Cys-SOH) [142,143]. This OhrR derivative remains bound to promoters until it undergoes further modifications into disulfide bond (Cys-S-S-R) or sulfenyl amide (Cys-SN) if reacting with a reduced cellular thiol or an amino group, respectively [144]. This results in the dissociation of OhrR from DNA and repression alleviation. The different OhrR cysteine derivatives can be reduced into thiol groups, allowing the regeneration of a reduced OhrR. Soonsanga et al. [145] also demonstrated the existence of an irreversible form of OhrR when the cysteine sulfenic acid derivative is further oxidized into cysteine sulfinic acid (Cys-SO_2_H). In that case, the cysteine sulfinic acid OhrR derivative is thought to be degraded.

It should be noted that another family of OhrR in which two cysteine residues are involved in organic hydroperoxides sensing has been described in *Xanthomonas campestris* [146]. In that case, the second cysteine residue provides the reduced thiol group to form the disulfide bond necessary to promote OhrR dissociation from DNA.

OhrR was first identified as a repressor of its own expression and of *ohrA*, which encodes a peroxiredoxin that scavenges organic peroxide [142,147,148]. Determination of the genome wide OhrR regulon performed in *Chromobacterium violaceum* indicated that the OhrR regulon encompasses, in fact, only a very limited number of genes in addition to *ohrA* and *ohrR*. In this bacterium, OhrR represses a putative diguanylate cyclase and activates indirectly three virulence-related genes [149]. *OhrR* mutants are generally more resistant to organic peroxide than their WT parental strain [150,151,152,153,154].

### 5.4. SoxRS

SoxRS control the expression of genes encoding defenses against superoxide. In *E. coli*, it is encoded by two adjacent and divergently transcribed genes, *soxR* and *soxS*. SoxR is a 17 kDa homodimer that belongs to the MerR transcriptional regulator family and SoxS is a 12 kDa transcriptional regulator of the AraC family. Each SoxR protomer contains an iron–sulfur cluster [2Fe-2S] that is oxidized by superoxide as well as by redox compounds [155,156]. The binding of oxidized SoxR to the *soxS* promoter activates *soxS* expression (Figure 6). Oxidized SoxR is reduced by NADPH-dependent Rsx/Rse enzymes [157].

In *E. coli*, genes activated by SoxS mainly encode factors that promote resistance against superoxide [158,159,160,161], including superoxide dismutases, YggG, a factor involved in iron–sulfur cluster repair [162], and the endonuclease IV involved in DNA repair [158]. The SoxRS regulon expression would return to a basal level upon proteolysis of SoxS [163].

The two proteins SoxRS are present together in proteobacteria and in actinomycetes [164]. Some bacteria, such as *Pseudomonas aeruginosa*, only contain SoxR and are devoid of SoxS. Other species, including the *Bacteroidetes Porphyromonas ginvivalis*, contain neither SoxR, nor SoxS.

### 5.5. Regulation of Oxidative Stress Defenses in Leptospira

A protein exhibiting the four characteristic cysteines and homology with typical SoxR is found only in some *Leptospira* spp. and not distributed widely in the whole genus (Figure 4). Identification of a canonical SoxS in *Leptospira* spp. is unclear. A protein exhibiting 33% of homology with *E. coli* SoxS has been found in *Leptospira* saprophytes (LEPBIa2624); however, this ORF contains extra domains that are not the landmarks of canonical SoxS. Whether LEPBIa2624-encoding regulator is a bona fide SoxS that controls the expression of *sodB* remains to be demonstrated. Neither SoxR, nor SoxS are found in *L. interrogans*.

Unlike most diderm bacteria, no OxyR has been reported so far in *L. interrogans*. A far ortholog of an OxyR-like LysR regulator has been annotated in *L. biflexa* (LEPBIa3010) and appears to be present in some but not all saprophytic species of *Leptospira*. However, there are no reports to date showing that it is a bona fide OxyR. Further studies are needed to determine whether this LysR regulator has a role in controlling the oxidative stress response in saprophytic *Leptospira* spp.

Two PerR-like regulators have been identified in *L. Interrogans*. PerRA (encoded by LIMLP_10155) was first identified as a Fur regulator whose inactivation elicits the up-regulation of catalase and peroxidase-encoding genes [165]. The second PerR, named PerRB (encoded by LIMLP_05620), was identified among the ORFs up-regulated when *L. interrogans* were exposed to H_2_O_2_ [80,166]. The distribution of PerRA and PerRB strikingly differs between saprophytes and pathogenic species. Homologs of PerRA and PerRB are present in all pathogenic species of the P1 clades sequenced so-far. However, PerRA is present in all saprophytes whereas homologs of PerRB cannot be found in non-pathogenic *Leptospira* species. Conversely, PerRB homologs are found in the P2 clade species where PerRA is generally missing [166,167] (Figure 4).

The respective function of PerRA and PerRB seems more complementary than redundant. Phenotypic and transcriptomic studies led to the conclusion that PerRA functions as a bona fide PerR in *L. interrogans. PerRA* inactivation leads to a higher survival in the presence of H_2_O_2_ than the WT [165,168]. Identification of the PerRA regulon indicates that not only PerRA represses the *ank-katE* operon (LIMLP_10145-10150) and genes encoding AhpC1 (LIMLP_05955) and CCP (LIMLP_02795), but it also activates a gene locus encoding a TonB-dependent transporter and the two-component system VicKR [165,166,167]. The main function of PerRA is therefore to control expression of defenses against H_2_O_2_.

The up-regulation of *perRB* in the presence of H_2_O_2_ suggests that, similarly to PerRA, PerRB is an ROS sensor. Indeed, classical PerRs, which generally self-repress their expression, dissociate from DNA in the presence of ROS, leading to their up-regulation. A *perRB* mutant has a comparable resistance to H_2_O_2_ as the WT strain. However, it exhibits a higher survival when exposed to paraquat [166]. From this phenotype, one can infer that *L. interrogans* PerRB represses genes involved in superoxide detoxification. RNASeq analyses of a *perRB* mutant when *L. interrogans* are cultivated in the EMJH culture medium or in dialysis membrane chamber did not provide an obvious explanation for this phenotype [166,167]. Indeed, the PerRB regulon encompasses either genes with unknown or poorly characterized function, or genes encoding factors involved in regulation (transcriptional regulators, c-di-GMP metabolism, and sigma factors). The putative mechanisms whose expression is controlled by PerRB and allowing to better tolerate deadly concentrations of superoxide are thus not understood. In the model that can be presently drawn, the main functions of PerRA and PerRB are to control expression of defenses against H_2_O_2_ and superoxide, respectively. As mentioned earlier, since no SoxRS system has been identified in *L. interrogans*, it is unknown how these bacteria orchestrate the response to superoxide. It is therefore tempting to speculate that PerRB fulfills the function of SoxRS in these bacterial species.

Interestingly, a limited number of genes, including the TonB-dependent transporter cluster, were found deregulated in both the *perRA* and *perRB* mutants, suggesting a certain level of redundancy in their regulon [166,167]. A double *perRAperRB* mutant is more resistant to both H_2_O_2_ and O_2_^•−^, indicating that the double mutant exhibits the respective phenotype of the single *perRA* and *perRB* mutants [166]. This double mutant was shown to be avirulent in the hamster and mice models and several virulence-related genes (including *ligA*, *ligB*, *lvrA*, *lvrB*, and *clpB*) were down-regulated in this mutant [166,167]. The exact role of PerRA and PerRB in the regulation of a virulence-associated network needs to be deciphered.

An ORF is annotated as a putative OhrR regulator in both pathogenic and saprophytes clades (LIMLP_17545 and LEPBI_I0798, respectively) (Figure 4) but no studies that characterized these proteins have been performed yet.

The transcriptional response to H_2_O_2_ has been characterized in *L. interrogans* [80]. Catalase and peroxidase-encoding genes (*katE*, *ccp* and *ahpC*) of the PerRA regulon were among the highest H_2_O_2_-responsive genes (i.e., genes rapidly up-regulated with sublethal doses of H_2_O_2_). Detoxification enzymes are therefore the first line of defense when pathogenic *Leptospira* are exposed to H_2_O_2_. The catalase is located in the periplasm in *L. interrogans* and is probably the main enzyme which allows a rapid elimination of H_2_O_2_ [70,80].

When *L. interrogans* are exposed to higher doses of H_2_O_2_, additional oxidative stress and redox-related genes are up-regulated including genes encoding thiol peroxidases, thioredoxin disulfide reductase, DsbD, Bcp, and Dps, as well as molecular chaperones from the heat shock response (DnaK/DnaJ/GrpE, GroEL/GroES, the small HSPs, and ClpB) and DNA repair proteins from the SOS response (RecA, RecN, DNA Pol IV, and LexA). Pathogenic *Leptospira* species have evolved to be very effective in rapidly breaking down H_2_O_2_ before it metabolizes into the very reactive ^•^OH. In addition, these bacteria are also well-equipped with a variety of repair mechanisms to heal oxidative damage to proteins and DNA occurring when H_2_O_2_ detoxifying enzymes are overwhelmed by the amount of ROS. Unlike what has been observed in other bacteria, genes encoding methionine sulfoxide reductases (that reduce methionine sulfoxide arising upon methionine oxidation), the iron–sulfur cluster synthesis and assembly SUF machinery, and the specific oxidative stress-related chaperone Hsp33 are not significantly up-regulated upon oxidative stress in *L. interrogans*, at least in the conditions that were tested.

## 6. Concluding Remarks and Perspectives

Understanding how *Leptospira* adapt to an oxidative environment, whether in the environmental niche or within the host, is far from being completely deciphered. The main limitation is the restricted number of laboratories that conduct research on *Leptospira* and leptospirosis, a neglected bacterium and disease. Another limitation is the difficulty to genetically manipulate these bacteria and to inactivate genes by allelic exchange, particularly in the pathogenic species. It is therefore not always possible to demonstrate the direct involvement of a particular factor in the oxidative stress response. Moreover, the mechanisms and pathways participating in the defense against oxidative stress are numerous and a certain level of redundancy and overlap is expected, whose study would require obtaining multiple mutants. The recent development of a new approach using the CRISPR/Cas9 system for inactivating genes in *Leptospira* [9,10,169] probably permits advancing our knowledge on the physiology of these bacteria and in particular on the different factors participating in the oxidative stress response. For instance, a mutant inactivated in *ccp* (LIMLP_02795), one of the most up-regulated ORF upon exposure of *L. interrogans* to H_2_O_2_, is not available. Determining the exact contribution of leptospiral CCPs in H_2_O_2_ detoxification would certainly benefit from having an effective method to generate mutants.

To date, most of our understanding of the oxidative stress response of *Leptospira* has focused on one particular ROS, H_2_O_2_, and how defenses against this oxidant are repressed by PerRA. Much less is known on how *Leptospira* adapt to the presence of other important ROS such as O_2_^•−^ or HOCl and what regulators control these adaptations, if any. In addition, important ROS detoxification machineries such as SOD/SOR are absent in pathogenic *Leptospira* species, which is unusual for aerobic bacteria. Do pathogenic *Leptospira* possess mechanisms to detoxify O_2_^•−^, a very important ROS produced by the host during infection, or do they rely exclusively on repair mechanisms against superoxide-mediated oxidative damage? Are *Leptospira* exposed to neutrophil-generated HOCl? These are among the important questions that would need to be answered in the future and require further investigation.

Pathogenic *Leptospira* and saprophytes have different repertoires of enzymes to detoxify ROS. Notably, pathogenic species have shown a strong catalase activity mediated by KatE, who is among the most expressed proteins in *L. interrogans*, while the saprophyte species did not have any detectable catalase activity even though their genome encodes a catalase of a different family (KatG). The specific role of KatG in the saprophytic species of *Leptospira* also remains to be elucidated.

In addition, as mentioned above, saprophytes possess an SOD, whereas pathogenic species do not. These observations indicate that *Leptospira* species might have evolved their antioxidant mechanisms according to their respective ecological niches and the nature of the ROS they are exposed to. Deciphering the evolution of the oxidative stress response within the different *Leptospira* clades will also be of great interest to reconstruct the evolutionary history of adaption to oxidative stress in this genus.

Finally, due to the importance of defenses against H_2_O_2_ for *Leptospira* virulence, it is tempting to propose that targeting the KatE catalase can be a successful therapeutic strategy. This requires to identify a specific inhibitor of the leptospiral KatE that does not affect the activity of human catalases, an important impediment in developing such drugs.

## Figures and Tables

**Figure 1 antioxidants-12-01273-f001:**
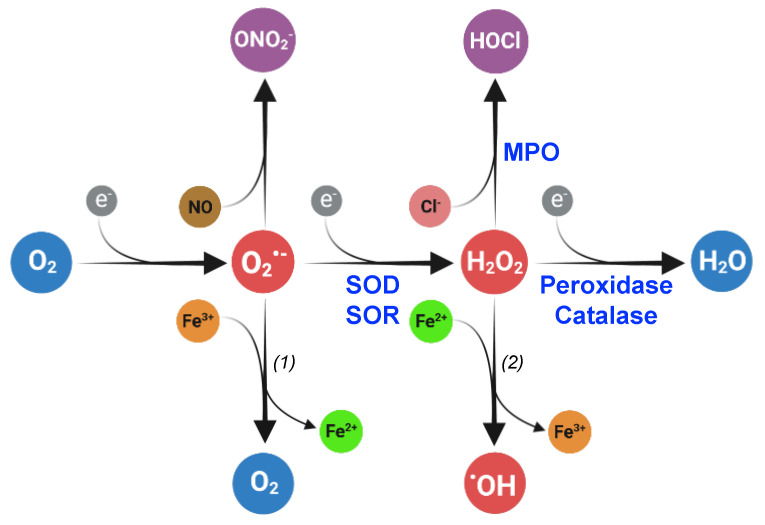
Superoxide (O_2_^•−^), hydrogen peroxide (H_2_O_2_), and hydroxyl radical (^•^OH) are reactive oxygen species (ROS) produced by the subsequent reduction in dioxygen (O_2_). In living organisms, the dismutation or reduction in superoxide, catalyzed by superoxide dismutase (SOD) and reductase (SOR), respectively, gives rise to hydrogen peroxide. Peroxidases and catalase catalyze the hydrogen peroxide reduction or dismutation, respectively, into water. Superoxide can react with nitric oxide (NO) to form peroxinitrite (ONO_2_^−^), and hydrogen peroxide can be transformed into hypochlorous acid (HOCl) in a reaction catalyzed by myeloperoxidases (MPO). The reactions (1) and (2) constitute the Haber–Weiss reaction that produces dioxygen and hydroxyl radical from superoxide and hydrogen peroxide, respectively. The reaction (2) is the Fenton reaction. Created with BioRender (BioRender.com).

**Figure 2 antioxidants-12-01273-f002:**
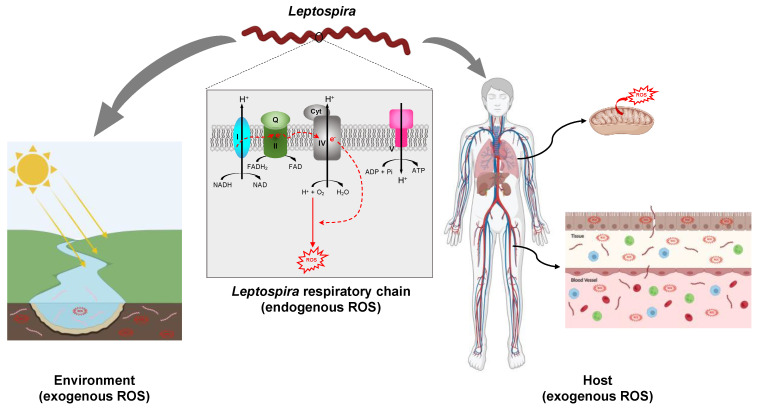
Exposition of *Leptospira* to ROS. *Leptospira* spp. are potentially exposed to ROS produced endogenously as a result of electron leakage from the respiratory chain. The putative complexes forming the respiratory chain in pathogenic *Leptospira*, complex I (encoded by LIMLP_00745-00750, LIMLP_03705-03760 in *L. interrogans* serovar Manilae), complex II (encoded by LIMLP_09980-09990 in *L. interrogans* serovar Manilae), complex IV (encoded by LIMLP_01080-01085, LIMLP_01375, LIMLP_01100-01110 in *L. interrogans* serovar Manilae), and complex V (encoded by LIMLP_06045-06080 in *L. interrogans* serovar Manilae) are represented. *Leptospira* spp. also face ROS produced in the environment (waters and soils). In addition, pathogenic species are exposed to ROS produced by the host phagocytic cells or tissues. The host mitochondria are one of the main ROS producers. Created with BioRender (BioRender.com).

**Figure 3 antioxidants-12-01273-f003:**
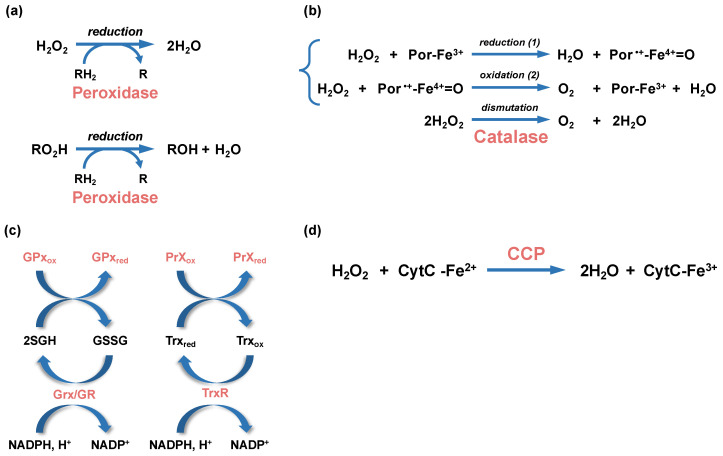
Enzymatic reactions leading to peroxide detoxification. (**a**) Hydrogen peroxide (H_2_O_2_) and alkyl peroxides (RO_2_H) are reduced by peroxidases (peroxiredoxin or glutathione peroxidases), with the participation of an organic electron donor (RH_2_). (**b**) The dismutation of H_2_O_2_ into water and dioxygen (O_2_) by heme-using catalases is a two-step reaction. In a first step (1), one molecule of H_2_O_2_ is reduced into water in a reaction where the heme ferric porphyrin (Por-Fe^3+^) acts as an electron donor and is oxidized into an oxoferryl porphyrin cation radical (Por^•+^-Fe^4+^=O). In a second step (2), a second molecule of H_2_O_2_ is oxidized into O_2_, allowing the regeneration of the ferric porphyrin. (**c**) Glutathione peroxidases (GPx) and glutaredoxin (Grx), which are oxidized during peroxide reduction or glutathionylation, use glutathione (GSH) as cofactors to regenerate their reduced forms, whereas the oxidized peroxiredoxin (PrX_ox_) is reduced by thioredoxin (Trx). The glutathione disulfide (GSSG) and oxidized thioredoxin (Trx_ox_) produced during these reactions are reduced by the glutathione reductase (GR) and thioredoxin reductase (TrxR), respectively. The NADPH/NADP^+^ is the redox couple involved in these reactions. (**d**) A simplified H_2_O_2_ reduction reaction by cytochrome C peroxidase (CCP) is represented. The CCP-bound heme is first reduced by the cytochrome C (CytC-Fe^2+^). The activated CCP reduces H_2_O_2_ into 2 molecules of H_2_O using the electron of its reduced heme.

**Figure 4 antioxidants-12-01273-f004:**
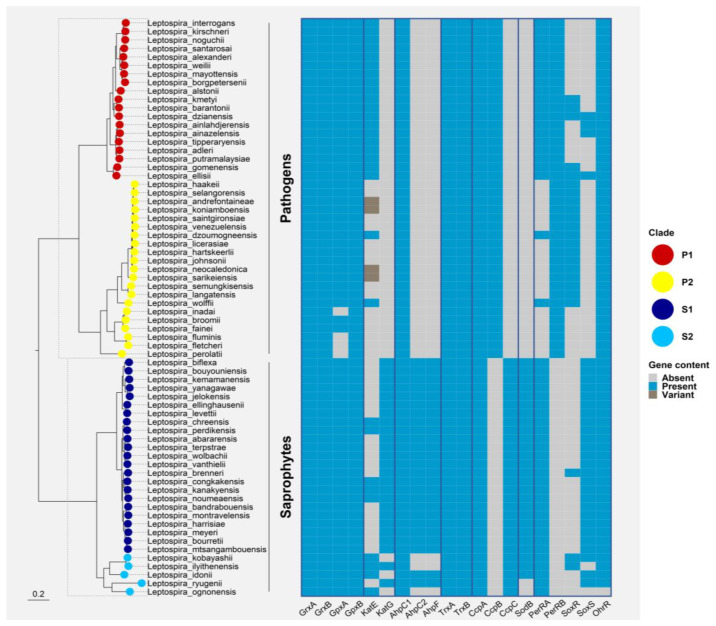
Phylogenetic distribution of ROS defenses and ROS-responsive regulators in all *Leptospira* spp. Ortholog sequences for each protein were searched with both BLAST v2.13.0 and HMMer v3.3.1 against a database of reference proteomes for the 68 *Leptospira* spp. with an e-value cut-off of 0.01 [72,73]. Alignments were performed with MAFFT v7.467 using the *L-INS-i* algorithm and tree inference was computed with FastTree v2.1.1 [74]. Direct homologs of the target protein were manually curated and selected from this phylogeny. These homologs were finally mapped to a core-genome tree of *Leptospirales* constructed based on a concatenated alignment of 1576 marker genes (identified with standalone OMA v2.5.0) and inferred with IQ-TREE v2.0.6 under the best-fit model of evolution (LG+F+R8) [75,76]. *Leptonema illini* and *Turneriella parva* were used as a root for the final *Leptospira* spp. phylogeny. The final figure was generated using the *ggtree* package for R [77]. Grx: glutaredoxins; GPx: glutathione peroxidase; KatE: catalase HPII; katG: catalase HPI; AhpC: Alkyl hydroperoxidase or Alkyl hydroperoxide reductase; AhpF: AhpC reductase; Trx: thioredoxin; Ccp: cytochrome C peroxidase; Sod: superoxide dismutase; PerR: peroxide stress regulator; SoxR: transcriptional activator and sensor of superoxide; SoxS: transcriptional activator of the SoxR regulon; OhrR: organic hydroperoxide resistance regulator.

**Figure 5 antioxidants-12-01273-f005:**
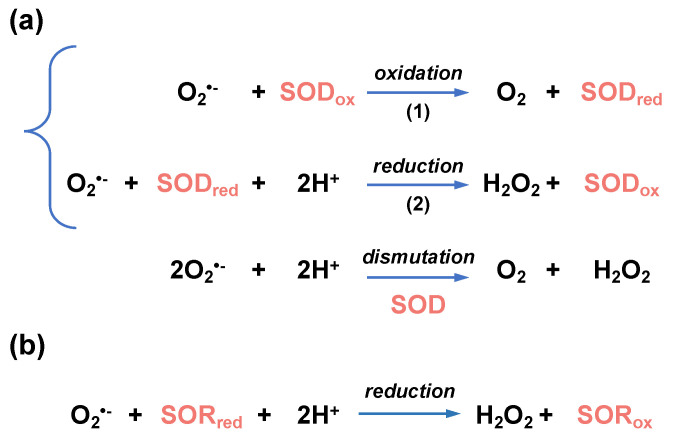
Enzymatic reactions leading to superoxide detoxification. (**a**) Superoxide dismutase (SOD)-catalyzed dismutation of O_2_^•−^. O_2_^•−^ is oxidized into O_2_ (reaction 1) and reduced into H_2_O_2_ (reaction 2). In these reactions, SOD is interconverted between an oxidized (SOD_ox_) and reduced (SOD_red_) forms. (**b**) Superoxide reductase (SOR)-catalyzed reduction in O_2_^•−^. During the reduction of O_2_^•−^ into H_2_O_2_, reduced SOR (SOR_red_) is converted into an oxidized form (SOR_ox_).

**Figure 6 antioxidants-12-01273-f006:**
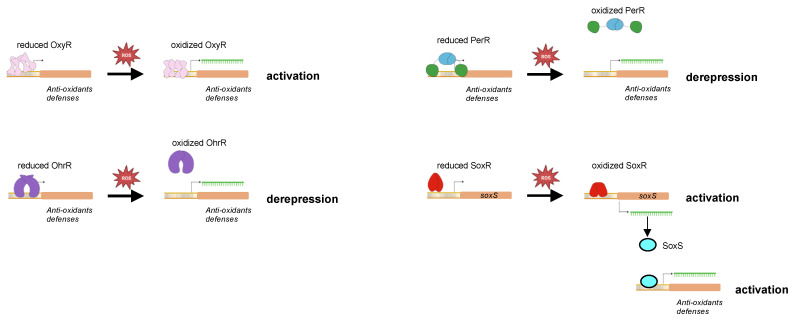
Oxidation-based regulation of DNA binding by oxidative-stress transcriptional regulators. The reduced and oxidized forms of different transcriptional regulators and ROS sensors are schematized in this cartoon. Oxidation of the regulators leads to conformational switch resulting either in a better DNA binding and activation of transcription (as for OxyR, SoxR, and SoxS) or in DNA dissociation and repression alleviation (as for PerR and OhrR).

## Data Availability

Data is contained within the article.

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
