# Peer review of "The Arsenal of Leptospira Species against Oxidants"

_antioxidants, 2023, doi:10.3390/antiox12061273_

Round 1

Reviewer 1 Report

This review paper is well written and introduced anti-oxidation mechanisms in genus Leptospira. Leptospira bacterium is classified into pathogens and saprophytes as non-pathogens.

Despite genetic tools were limited, recently some research groups reported various aspects in disease control and basic science in Leptospira. Here the authors explain oxidative stress response systems in Leptospira based on enzymatic and non-enzymatic defense, and anti-oxidation response transcription factors. This knowledge purchase to us the expanded information and scientific understanding.

Nevertheless, many commonly known concepts and principles, especially ROS reactions and antioxidant, are related to bacterial defense systems. Therefore, I suggested that MS needs to be shortened and fine-tuned. If the MS is correctly modified, it can be published in antioxidants to support interesting information for many readers.

Minor comments

1.     Fig. 1. Maybe Haber-Weiss reaction involved Fenton reaction, especial Fig. 1 reaction number 2.

2.     Section 2 is a general info. It does not significant.

3.     Line 225. Need to add the examples for thiol-redox buffers as low molecular weight molecules. Nevertheless, in section 4.2 explained thiol-redox buffers contained molecules. How about bacilithiol (BSH) and mycothiol (MSH) in genus Leptospira? Are both LMW thiols not produced or genes missed in genomes?

4.     Fig. 3. Need to size up!

5.     Line 381. GR is glutathione reductase?

6.     Section 5. Maybe need to focus into Leptospira.

7.     Line 504. Correctly revised species name for Neisseria.

8.     Line 298, 546. Please write full name of bacterial species.

9.     Lines 547-548. Maybe delete!

10.  Line 556. Maybe remove! OxyR gene not found in Genus Leptospira genomes till now? Need to write clearly.

11.  Line 586. The ci-diGMP revised c-diGMP. C is cyclic means?

12.  Lines 598-600. Might be change the sentence clearly. PerRA-RB double mutants of Leptospira testified into hamster and mice, right? How about the bacterial infection virulence between WT and double mutant?

13.  Revised references followed journal guideline.

Author Response

Response to reviewer 1 comments

We would like to thank the reviewer for his(her) very interesting, pertinent, and constructive comments and suggestions.

 Point 1: Fig. 1. Maybe Haber-Weiss reaction involved Fentonreaction, especial Fig. 1 reaction number 2.

 Response 1. We believe that the reviewer is suggesting clarifying that the Fenton reaction is the second step of the Haber-Weiss reaction. We have therefore clarified this issue in the section 2 (lines 120-121) and in the legend of Figure 1 (lines 98-100).

Point 2: Section 2 is a general info. It does not significant.

 Response 2. We totally agree with the reviewer that the section 2 describes “many commonly known concepts and principles” on “ROS reactions and antioxidants”. This may seem unnecessary but, in fact, it was intentional. From many discussions we have had with the Leptospira community, and the spirochetes community in general, we realized that many spirochetologists are not familiar with the general principles of the oxidative stress response in bacteria. Therefore, we believe that it is important to give a general overview of oxidants and antioxidant defenses that will be useful for an audience which is not well-acquainted with the biology of the oxidative stress.

Point 3: Line 225. Need to add the examples for thiol-redoxbuffers as low molecular weight molecules. Nevertheless, in section 4.2 explained thiol-redoxbuffers contained molecules. How about bacilithiol(BSH) and mycothiol (MSH) in genus Leptospira? Are both LMW thiols not produced or genes missed in genomes?

 Response 3. Bacillithiol biosynthesis classically requires three genes (bshABC). Although some Leptospira encode for a far homolog of BshB, no BshA nor BshC could be clearly identified. Mycothiol biosynthesis involves four genes (mshABCD). There is a far homolog of MshB in some Leptospira spp., but mshD, the last gene of the biosynthesis pathway, is missing. Therefore, there is no evidence that bacillithiol and mycothiol are synthesized in Leptospira and that these molecules play a role as redox buffers in Leptospira spp. To avoid increasing the length of the review, this point is not included in the review.

Point 4: Fig. 3. Need to size up!

Response 4. As suggested by the reviewer, we have increased the size of the elements of Figure 3.

Point 5: Line 381. GR is glutathione reductase?

Response 5. GR stands for glutathione reductase. “Glutathione reductase” term is already written out in full and the GR abbreviation initially used in line 259 (formally line 241).

Point 6: Section 5. Maybe need to focus into Leptospira

Response 6. For the same reason developed in point #2, we believe that it is important to properly introduce the features of the main oxidative stress-related transcriptional factors for readers not familiar with these regulators. This allows a better comprehension of the specificity of the oxidative stress response of Leptospira.

Point 7: Line 504. Correctly revised species name for Neisseria.

Response 7. We thank the reviewer for detecting this mistake and this was corrected (line 517).

Point 8: Line 298, 546. Please write full name of bacterial species.

Response 8. We thank the reviewer for detecting these mistakes and they were corrected (lines 308 and 557).

Point 9: Lines 547-548. Maybe delete!

Response 9. As suggested by the reviewer, we have deleted part of the sentence. The other part has been fused with another sentence (lines 547-548).

Point 10: Line 556. Maybe remove! OxyR gene not found in Genus Leptospira genomes till now? Need to write clearly.

Response 10. We believe that emphasizing that pathogenic Leptospira are devoid of SoxRS is as important as the fact that they lack any SOD/SOR. Therefore, we have maintained the sentence originally located at line 556 (now at line 574).

A LysR transcriptional factor has been annotated as putative OxyR-like only in some saprophytic Leptospira genomes. Further phylogenetic and functional studies will be necessary to establish the role of this transcriptional regulators and to demonstrate whether it functions as a bona fide OxyR. As suggested by the reviewer, we have commented on the putative presence of OxyR across the Leptospira genus in section 5.5 (lines 575-580).       

Point 11: Line 586. The ci-diGMP revised c-diGMP. C is cyclicmeans?

Response 11. We thank the reviewer for noticing this mistake and this was corrected (line 609).

Point 12: Lines 598-600. Might be change the sentence clearly.PerRA-RB double mutants of Leptospira testified intohamster and mice, right? How about the bacterialinfection virulence between WT and double mutant?

Response 12. The virulence of the double perRAperRB mutant has been tested in hamster and mice and the double mutant has lost its virulence in both animal models, and unlike the WT strain L. interrogans, the double mutant cannot colonize hamster nor mice kidney and liver. This is already mentioned in the manuscript  : “This double mutant was shown to be avirulent in the hamster and mice models and several virulence-related genes (including ligA, ligB, lvrA, lvrB, clpB) were down-regulated in this mutant [167,168]. The exact role of PerRA and PerRB in the regulation of a virulence-associated network needs to be deciphered.” (lines 621-625).

Point 13: Revised references followed journal guideline.

Response 13. We have checked that the references were formatted according to the Antioxidants recommended style. We have also verified that all the references were complete and have corrected references 4 (lines 714-715), 33 (line 777), and 113 (line 968) that were formatted incompletely.  

Reviewer 2 Report

Review of the paper entitled „The arsenal of Leptospira species against oxidants” by Samuel G. Huete and Nadia Benaroudj

      The aim of the Authors was to describe the reactive oxygen species (ROS) encountered by Leptospira species in their different ecological niches and outline the repertoire of defense mechanisms identified so far to scavenge deadly for these bacteria ROS. The Authors also point out that all leptospires are exposed to ROS in the environment but only pathogenic species are well-equipped to sustain the oxidative stress encountered inside their hosts during infection. This is an interesting paper.

 My comment

     Knowledge of the defense mechanisms that protect the pathogenic bacteria against environmental factors harmful to them is important for the design of antibacterial drugs used in practical clinical medicine. In the light of the knowledge presented by the Authors, it seems that the currently "popular" antioxidants should not be used to treat leptospirosis. However, it would be beneficial to use drugs that increase the level of ROS. Are such studies conducted? What are the results of these studies? It would be good for the Authors to expand their paper with this type of information and briefly discussed this issue.

Author Response

Response to reviewer 2 comments

We would like to thank the reviewer for his(her) very interesting, pertinent, and constructive comments and suggestions.

Point 1: Knowledge of the defense mechanisms that protect the pathogenic bacteria against environmental factors harmful to them is important for the design of antibacterial drugs used in practical clinical medicine. In the light of the knowledge presented by the Authors, it seems that the currently "popular" antioxidants should not be used to treat leptospirosis. However, it would be beneficial to use drugs that increase the level of ROS. Are such studies conducted? What are the results of these studies? It would be good for the Authors to expand their paper with this type of information and briefly discussed this issue.

Response 1. To the best of our knowledge, the use of host-directed drugs that increase ROS production by the host ­­–and thereby could have a bactericidal activity against Leptospira ­­– has never been investigated for leptospirosis. Another therapeutic strategy would be to target the catalase KatE, whose inactivation attenuates Leptospira virulence. A specific inhibitor of the leptospiral KatE that does not impact the activity of mammal catalase would be necessary. As suggested by the reviewer, we have commented on this at the end of the manuscript (lines 695-699).       

Round 2

Reviewer 1 Report

The revised manuscript has been well organized and the contents have become clear.
The request for a brief description of the general oxidative stress response has not been modified, but it does not matter if it is unavoidable for an overall explanation.
The reference part will still need more attention. Because the name or short name of the journal should be corrected properly. I hope that this review will increase the direction and cooperation of research on oxidative stress response to Leptospira.